# Impact of Enzymatic Hydrolysis and Microfluidization on the Techno-Functionality of Oat Bran in Suspension and Acid Milk Gel Models

**DOI:** 10.3390/foods11020228

**Published:** 2022-01-15

**Authors:** Natalia Rosa-Sibakov, Maria Julia de Oliveira Carvalho, Martina Lille, Emilia Nordlund

**Affiliations:** VTT Technical Research Centre of Finland Ltd., P.O. Box 1000, FI-02044 Espoo, Finland; maria.julia.carv@gmail.com (M.J.d.O.C.); Martina.Lille@vtt.fi (M.L.); emilia.nordlund@vtt.fi (E.N.)

**Keywords:** oat bran, β-glucan, β-glucanase, mechanical treatment, acid milk gel

## Abstract

Oat bran is a nutritionally rich ingredient, but it is underutilized in semi-moist and liquid foods due to technological issues such as high viscosity and sliminess. The aim of this work was to improve the technological properties of oat bran concentrate (OBC) in high-moisture food applications by enzymatic and mechanical treatments. OBC was hydrolyzed with β-glucanase (OBC-Hyd) and the water-soluble fraction (OBC-Sol) was separated. OBC, OBC-Hyd and OBC-Sol were further microfluidized at 5% dry matter content. Enzymatic treatment and microfluidization of OBC reduced the molecular weight (Mw) of β-glucan from 2748 kDa to 893 and 350 kDa, respectively, as well as the average particle size of OBC (3.4 and 35 times, respectively). Both treatments increased the extractability of the soluble compounds from the OBC samples (up to 80%) and affected their water retention capacity. OBC in suspension had very high viscosity (969 mPa·s) when heated, which decreased after both enzyme and microfluidization treatments. The colloidal stability of the OBC in suspension was improved, especially after microfluidization. The addition of OBC samples to acid milk gels decreased syneresis, improved the water holding capacity and softened the texture. The changes in the suspension and gel characteristics were linked with reduced β-glucan Mw and OBC particle size.

## 1. Introduction

Oats are increasingly used in various food products due to health-promoting and sustainability reasons. Oats contain many beneficial components, such as dietary fiber (DF), protein and minerals [1]. Mixed-linkage (1→3), (1→4)-β-d-glucan (β-glucan), and arabinoxylan are the main DF components of oats. The European Food Safety Authority has authorized health claims for β-glucan on the reduction of blood plasma cholesterol concentrations and the attenuating post-prandial glycemic response [2]. The recommended daily intake is 3 g of oat β-glucan, and the claim can be used for food products containing at least 1 g per portion. Even though oats, especially bran, is widely used in various cereal food products, such as porridges, breads, snacks and biscuits [3,4,5], it can be challenging for consumers to reach the recommended intake of β-glucan. Beverages and spoonable products such as yogurts are also potential food categories for addition of high-DF oat ingredients [6]. However, the incorporation of oat ingredients into semi-moist and liquid food products at the amounts needed to reach β-glucan claims is challenging due to the increase of viscosity and sliminess by β-glucan, and coarse mouthfeel [7].

In the liquid category, the best-known oat-rich product is an oat-based milk substitute, also called “oat milk.” However, the DF content of oat milks is usually very low (~0.8 g DF per 100 mL in commercial products) because β-glucan increases the viscosity of the product, and therefore, a major part of the DF is typically either depolymerized or filtered away [8]. In drinkable applications, the use of β-glucan with high molecular weight (Mw) is very difficult because it tends to form semi-solid or concentrated dispersions, resulting in increased viscosity. In order to obtain a suspension with high concentration of β-glucan and low viscosity, the Mw of β-glucan must be reduced by, for example, acid or enzymatic depolymerization [9], adding ascorbic acid [10] or high-pressure homogenization [11]. Microfluidization is a mechanical treatment that pumps a suspension of particles at high pressures through interaction chambers, where the particle size is reduced by impact and very high shear rates [12]. Although drinkable applications require a reduction of the Mw of β-glucan, it is worth mentioning that the physiological functionality of β-glucan has also been linked to its Mw. In our previous studies, we observed that extensive depolymerization of β-glucan (<400 kDa) should be avoided in order to retain its capability to retain bile acids and increase the intestinal viscosity in vitro [13] and in vivo [14]. On the other hand, when subjects consumed a meal with low β-glucan Mw (82 kDa), a higher amount of phenolic compounds was detected in urine compared to medium and high Mw (524 and 1000 kDa, respectively) [14]. Therefore, both the techno-functional and physiological properties of β-glucan should be taken into consideration during the processing of oat-based foods.

The addition of purified β-glucan to dairy matrices and its impact on their physical and textural attributes have been studied by many authors [15,16,17,18]. Mixtures of high-Mw β-glucan and milk proteins at sufficient levels to reach health claims have shown undesirable appearance and texture due to thermodynamic incompatibility and phase separation [19,20]. Incorporation of β-glucan into acid milk gels or yogurts has been shown to alter the gelation process by interfering in the formation of the casein network and resulting in significantly weaker gels [16,17]. However, β-glucan may compensate for this interference by forming an alternative, water-entrapping secondary gel structure within the protein network, with long polysaccharides among casein micelles or small spherical clusters within the yogurt matrix [21,22,23]. The extent to which β-glucan can improve or degenerate the yogurt structure seems to be largely dependent on the source of origin, the extraction and purification methods of the β-glucan-rich ingredient, as well as β-glucan Mw and concentration. Most of the studies found in the literature were performed with highly purified β-glucan ingredients [15,16,17,18]. Purification processes can lead to changes in the Mw of β-glucan and eliminate other bioactive compounds present in cereal fractions, and this is not feasible in terms of industrial processes. Therefore, it is important to perform studies with ingredients which contain sufficient β-glucan for the final products to meet health claim requirements, such as oat bran.

The aim of this work was to investigate the enzymatic hydrolysis of oat bran concentrate (OBC) with β-glucanase, physical treatment of OBC by microfluidization, and the combination of the enzymatic and physical treatments for modification of the physicochemical properties of OBC in an attempt to improve OBC behavior in high-moisture food applications.

## 2. Materials and Methods

### 2.1. Raw Material and Enzyme

Oat bran concentrate (OBC) was obtained from Raisio Group (Raisio, Finland). The OBC contained 21.1 g of β-glucan in 100 g d.m., determined by a standard method 32–23.01 (AACC, 2000) using the Megazyme β-glucan mixed-linkage assay kit (Megazyme International Ireland Ltd., Wicklow, Ireland). Commercial enzyme preparation (Depol 740 L) was obtained from Biocatalyst Ltd., Cardiff, UK. This enzyme had β-glucanase activity of 9888 nkat/mL when assayed, as described by Bailey and Nevalainen [24] using barley β-glucan (Megazyme) as substrate instead of hydroxyethyl cellulose.

### 2.2. Study Design

The experimental setup of this study is illustrated in Figure 1. OBC was treated with β-glucanase (OBC-Hyd), microfluidization (OBC-M), and their combination (OBC-Hyd-M). Enzymatic treatment was also used to prepare a soluble OBC fraction (OBC-Sol), which has previously shown potential for drinkable applications [25], and this fraction was also further microfluidized (OBC-Sol-M). The impact of the different treatments was studied by analyzing OBC characteristics such as β-glucan Mw, particle size, and water retention capacity, as well as the viscosity and colloidal stability of the OBC samples in suspension. Subsequently, the applicability of the OBC samples in a dairy matrix was evaluated by preparing acid milk gels with the OBC amounts necessary to reach β-glucan health claims. The quality of acid milk gels enriched with oat samples was evaluated by analyzing the gel texture and its water retention ability.

### 2.3. Enzymatic Treatment and Microfluidization

OBC was enzymatically treated based on the process described by Aktas et al. [25] with few modifications. OBC was first preconditioned (30% moisture content) after which OBC and the enzyme solution (50 nkat β-glucanase activity/g dry OBC) was fed into a twin-screw extruder (APV MPF 19/25, Baker Perkins Ltd., Peterborough, UK) at 50 °C. After the extrusion phase, the formed dough-like OBC mass was incubated in a closed vessel at 50 °C for 1 h. The enzymatic treatment was stopped by feeding the OBC sample through the same extruder at 110 °C. After the enzyme inactivation, the material was dried in a conventional oven (50 °C) overnight and finely ground using an impact pin-disc mill 100 UPZ (Hosokawa Alpine, Augsburg, Germany). This sample was named OBC-Hyd. A water-soluble fraction, referred to as OBC-Sol, was prepared with the same procedure described above, but the enzyme inactivation was performed by adding boiling water (100 °C) to the vessel containing the incubated OBC sample. The mixture was stirred with a homogenizer (Heidolph Diax900 Ultra Turrax, Gemini BV, Apeldoorn, The Netherlands) at 12,000 rpm for 2 min. After that, the sample was centrifuged (15 min, 6100× *g*) and the supernatant was freeze-dried in an Epsilon 2–25 freeze-dryer (Martin Christ Gefriertrocknungsanlagen GmbH, Osterode am Harz, Germany) with a standard program and ground with a cutting mill.

The two enzymatically treated OBC samples, OBC-Hyd and OBC-Sol, as well as the native OBC sample were further microfluidized according to Rosa-Sibakov et al. [26]. Briefly, the oat samples were mixed with distilled water at 5% *w/w*. The microfluidizer (Microfluidics International Corporation, Westwood, MA, USA) chambers had a diameter of 400 μm during the first pass, 400 μm and 200 μm during the second pass, and 400 μm and 100 μm during the third and fourth passes. The operating pressure was 1850 bar. The microfluidized samples were named OBC-M, OBC-Hyd-M and OBC-Sol-M. Sodium azide (0.01% *w/w*) was added to avoid microbial growth and the samples were stored at 4 °C until further analysis.

### 2.4. β-Glucan Molecular Weight (Mw) Distribution

The Mw distribution of β-glucans in oat samples was determined by high-performance size-exclusion chromatography (HP SEC), which consisted of an Alliance 2690 separation module, using Calcofluor white (Fluorescent Brightener 28, Aldrich, Germany) and Scanning Fluorescence 474 detector (Waters Inc., Milford, MA, USA), as described in Suortti [27]. The samples were prepared by dissolving 20 mg into 10 mL of NaOH (0.1 M) and 0.1% NaBH_4_ under magnetic stirring at room temperature overnight. The samples were diluted and filtered using a 0.45 µm syringe filter prior to HP SEC measurement. The linear size-exclusion calibration curve was prepared with β-glucan standards from Megazyme ranging from 33.6 to 667 kDa.

### 2.5. Preparation of the OBC Suspensions

For the following characterizations (described in Section 2.6, Section 2.7, Section 2.8 and Section 2.9), the dry samples (OBC, OBC-Hyd and OBC-Sol) were dispersed in water at 5% *w/w* by magnetic stirring (25 °C, 1 h) before analysis. The microfluidized samples (suspensions) were analyzed as is (5% *w/w*).

### 2.6. Particle Size

The particle size distribution of OBC samples was measured by using a laser diffraction particle size analyzer, Mastersizer 3000 (Malvern Instruments Ltd., Worcestershire, UK). The suspensions (5% *w/w*) were analyzed in the liquid module when the obscuration was between 2 and 15%. Distilled water was used as carrier and data were calculated using Fraunhofer approximation. At least three parallel analyses were carried out per sample.

### 2.7. Microscopy

Phase contrast microscopy images of OBC sample suspensions (5% *w/w*) were obtained with a light microscope, Olympus BX-40 (Olympus Corp., Tokyo, Japan). A drop of sample was placed on a glass slide and spread to a thin layer by pressing a cover glass on top of it. The samples were observed as such in the microscope, without any staining, with a 10× objective.

### 2.8. Water Retention Capacity (WRC) and Extractability

Total water retention capacity (WRC) and extractability of OBC samples were determined in triplicate based on the method described by De Bondt et al. [28] with some modifications. Briefly, OBC suspensions (5% *w/w*) were centrifuged for 25 min at 4000× *g*, after which the supernatant was separated from the pellet by decantation. The pellet was weighed, and the supernatant and pellet were dried (90 °C overnight). The extractability was calculated based on the dry weight of the soluble mass present in the supernatant and the insoluble pellet. The WRC was expressed as the amount of water (mL) held by the mass of oat sample (g), according to Equation (1): (1)$$\mathrm{WRC}=\frac{\mathrm{mass}\;\mathrm{pellet}\;\mathrm{after}\;\mathrm{centrifugation}-\mathrm{mass}\;\mathrm{dry}\;\mathrm{pellet}-\mathrm{mass}\;\mathrm{dry}\;\mathrm{supernatant}}{\mathrm{mass}\;\mathrm{dry}\;\mathrm{pellet}+\mathrm{mass}\;\mathrm{dry}\;\mathrm{supernatant}}$$

### 2.9. Colloidal Stability

The OBC suspensions (20 mL) were transferred to graduated 25 mL tubes in triplicate. The volume of sediment was measured visually at different time points (0, 1, 4 and 24 h) at room temperature. The results are the average of four different observations. The colloidal stability was expressed as the volume of the serum phase, according to Equation (2):Colloidal stability (mL) = initial volume of the suspension − volume of the sediment (2)

### 2.10. Viscosity

The viscosity of OBC suspensions was analyzed with a Rapid Visco Analyser (RVA Super 4, Newport Scientific, Warriewood, NSW, Australia). All OBC samples were measured as aqueous suspensions at a solid content resulting in 1% β-glucan in the mixture (dry matter content varied depending on the β-glucan content of each OBC sample). The weight of the measured suspension samples was 25 g. The dry OBC samples (OBC, OBC-Hyd and OBC-Sol) were prepared for the measurement by weighing the required amount of OBC sample and water into the RVA cup just before starting the measurement. The microfluidized OBC samples were measured right after microfluidization as such (OBC-M and OBC-Hyd-M) or after diluting the sample with water to a β-glucan content of 1% (OBC-Sol-M). The RVA measurement consisted of the following steps: (1) 30 s at 50 °C, (2) 60 min at 50 °C, (3) heating to 90 °C at 10 °C/min, (4) 5 min at 90 °C, (5) cooling to 20 °C at 10 °C/min, and (6) 5 min at 20 °C. The sample was continuously mixed during the measurement. The mixing speed was 960 rpm during step 1 and 160 rpm during all other steps. As a result, the viscosities after 60 min at 50 °C (end of step 2) and the final viscosity after heating and cooling to 20 °C (end of step 6) were reported. The measurements were carried out in triplicate.

### 2.11. Preparation of Acid Milk Gels Enriched with Oat Samples

Acid milk gels were formulated to contain 9.6% (*w/w*) milk solids (i.e., 3.3% milk protein) and 1 g of β-glucan per portion (portion size 150 g) in order to achieve the level of β-glucan required for health claims. OBC samples were mixed with distilled water by constant magnetic stirring (200 rpm) at 50 °C for 30 min. Skimmed milk powder (Valio Ltd., Helsinki, Finland) was reconstituted by adding the milk powder to the OBC-water mixture, and mixed for another 30 min by constant magnetic stirring at 50 °C. The new mixture (reconstituted skim milk + OBC samples) was heat-treated (90 °C for 5 min) and cooled down in ice to 40 °C. Glucono-δ-lactone (Sigma Chemicals, St. Louis, MO, USA) was added to the mixture at 1.3% (*w/w*) and mixing was continued for 5 min. After that, the mixture was poured into plastic cups, centrifuge tubes and glass slides for further analysis (texture, water holding capacity and microstructure, respectively) as described below. The samples were incubated at 40 °C until the pH reached 4.5–4.6 (around 4–5 h). The resulting gels were characterized straight after acidification. For texture measurement, additional gels were prepared and stored one week at 4 °C. For these later gels, sodium azide (0.01% *w/w*) was added just before the acidification procedure in order to prevent microbial growth. Control samples were prepared in a similar manner without OBC. The acid milk gels were prepared in triplicate in two different days (*n* = 6).

#### Characterization of Acid Milk Gels Enriched with OBC Samples

For measuring the spontaneous syneresis, the acid gels (40 g) were formed in tightly sealed cups (ø 3.5 cm) and the water released from the top of the gel at the end of incubation period was carefully removed and weighed. After removing the water released from the gel, the firmness of the gels was measured with a large deformation test using a TA.XTPlus Texture Analyzer (TA, Stable Microsystems Ltd., Godalming, UK) equipped with a 5 kg load cell. Gel deformation was done by penetrating the gel with a ø 25 mm cylinder probe at a constant speed of 1.0 mm/s to a distance of 75% of the sample height. The area under the force vs. distance curve (up to 15 mm) was determined and taken as a measure of gel firmness. Deformation tests were performed on six individual gels straight after acidification or after one week storage at 4 °C.

For the measurement of water holding capacity, the acid gels (20 g) were formed in 50 mL centrifuge tubes. After acidification, the tubes were placed in ice, centrifuged (3000× *g*, 15 min, 4 °C) and the supernatant was drained carefully and weighed [29]. The acid milk gels were prepared in triplicate on two different days (*n* = 6) and the water holding capacity was expressed as the percentage of the weight of the pellet in relation to the initial weight of the gel, according to Equation (3):(3)$$\mathrm{Water}\;\mathrm{holding}\;\mathrm{capacity}\;\left(\%\right)=\frac{\mathrm{mass}\;\mathrm{pellet}\;\mathrm{after}\;\mathrm{centrifugation}}{\mathrm{mass}\;\mathrm{of}\;\mathrm{the}\;\mathrm{gel}}\times 100$$

Gel microstructure was examined with a confocal laser scanning microscope (CLSM; Bio-Rad, Hemel Hempstead, Herts, UK) using Calcofluor White (Fluorescent Brightener 28, Sigma-Aldrich, Steinheim, Germany) and Rhodamine B (Rhodamine B amine, Sigma-Aldrich) to stain the glucan containing cell wall structures and the proteins, respectively, as described by Kortekangas et al. [30]. 

### 2.12. Statistical Analysis

Statistical analysis was performed using IBM SPSS Statistics 21 (IBM Corporation, New York, NY, USA) based on one-way ANOVA followed by a Tukey’s test at 95% confidence level (*p* < 0.05). Correlation coefficients (r) were carried out by Pearson’s correlation method with a 95% confidence interval.

## 3. Results and Discussion

### 3.1. Impact of Processing on OBC Characteristics

The enzyme treatment of OBC reduced the Mw of β-glucan by 3-times (893 kDa, OBC-Hyd) when compared to native OBC (2748 kDa) (Table 1). The Mw distribution of β-glucan in OBC-Hyd had clearly higher dispersity (6.7) than OBC (2.1) and a wider distribution with two peaks (Figure 2a). The OBC-Sol sample, representing the water-soluble fraction after the enzyme treatment, contained the highest amount of β-glucan (44%) with clearly lower Mw (294 kDa) than that of OBC or OBC-Hyd. By using a similar enzymatic process, Aktas et al. [25] prepared a soluble β-glucan fraction (44%) with much lower Mw (11 kDa). However, they used a longer incubation time (4 h), whereas intendedly shorter incubation time (1 h) was used in the present study to have a higher Mw in the OBC-Sol sample in an attempt to better retain physiological functionality of β-glucan [13].

Microfluidization of the OBC samples significantly reduced β-glucan Mw, also after the enzyme treatment (Table 1). The biggest Mw drop was observed in microfluidized native OBC (Mw reduced from 2748 to 350 kDa). The β-glucan content in microfluidized samples was considered the same as in the corresponding OBC samples, as no degradation of β-glucan was expected during microfluidization. The β-glucan Mw distribution of the microfluidized samples was more homogeneous with single peaks when compared with the corresponding enzyme-treated OBC samples (Figure 2a). The mechanical forces applied to OBC during microfluidization were, therefore, much more effective to reduce β-glucan Mw than the enzymatic treatment with a β-glucanase. Kivelä et al. [11] also observed a decrease in the Mw of oat β-glucan from 1440 kDa down to 130 kDa, when microfluidization was applied to a water-extracted dispersion from oat bran containing 0.15% β-glucan. The present study showed that microfluidization is an efficient method to degrade β-glucan when it is present in a complex matrix of oat bran interlinked with other cell wall polymers and compounds. Moreover, the microfluidization was performed at an OBC concentration of 5% (i.e., 1.1, 1.1 and 2.2% of β-glucan for OBC-M, OBC-Hyd-M and OBC-Sol-M, respectively), which is about 10 times more than the concentration used by Kivelä et al. [11].

A positive correlation was found between the β-glucan Mw and the average particle size (d_50_) of OBC samples (r = 0.99 **). The OBC had an average particle size (d_50_) of 426 µm (Table 1) and a volume distribution with one sharp single peak (Figure 2b). The particle size was reduced to 124 µm in the OBC-Hyd, which was due to the process including enzyme treatment in the extruder and further grinding after drying. OBC had an intact structure of the outer layers, such as pericarp and rows of aleurone layers (Figure 3a, indicated by the arrows). These structures were still visible in OBC-Hyd (Figure 3b), but with a clearly smaller size and with the rows of aleurone cells less apparent. OBC-Sol had the smallest particle size (d_50_ = 14 µm), as expected for this soluble fraction, and only very small particles (Figure 3c). Microfluidization drastically reduced the particle size of the OBC samples down to 6–12 µm, as previously observed for wheat bran [26,28,31]. The microscopic images of the microfluidized OBC samples (Figure 3d–f) show that the cell wall structures were all broken down and a more homogeneous and uniform microstructure was achieved. All the microfluidized samples in the present work had a bi-modal particle size distribution (Figure 2b). De Bondt et al. [28] described that microfluidization of wheat bran reduced its particle size to 14.8 μm with a unimodal distribution. The bi-modal distribution in microfluidized OBC samples suggests re-aggregation of the β-glucan molecules with small Mw [32] or expansion of the particles [31]. Comparing the microscopic pictures of OBC-Sol and OBC-Sol-M, the microfluidized OBC sample had clearly more aggregates.

The WRC of OBC and OBC-Hyd were at similar levels (3.0–3.1 mL/g), and microfluidization reduced their WRC to 2.5 and 2.3 mL/g, respectively, for OBC-M and OBC-Hyd-M. Microfluidization has been shown to increase the WRC of wheat bran [28,31] and insoluble oat fraction [33], which is the opposite behavior observed in the OBC-M and OBC-Hyd-M. The differing results are probably due to the differences in the raw materials; OBC has more soluble fractions compared to highly insoluble wheat bran and the insoluble oat fraction studied by Valoppi et al. [33]. Thus, in the present study, microfluidization caused further degradation and subsequent solubilization of DF components and other bran components in the OBC samples with reduced WRC, whereas in the previous work a decrease in the particle size increased the surface area of particles to bind water, but the particle components did not yet solubilize. The enhanced solubilization of OBC components was also supported by increased extractability values in OBC-M and OBC-Hyd-M (2.7 and 2.2-times) when compared to OBC and OBC-Hyd, respectively (Table 1).

However, microfluidization of OBC-Sol, which expectedly had the lowest WRC (1.9 mL/g), had an opposite effect compared to other samples. In this case, the microfluidization increased the WRC by 3-fold (1.9 and 5.8 mL/g for OBC-Sol and OBC-Sol-M, respectively). Interestingly, microfluidization reduced the extractability of this OBC-Sol-M (from 80 to 39%). In fact, a negative correlation was found between the WRC and extractability of the OBC samples (r = 0.93 *) when the OBC-Sol-M is excluded from the analysis. This suggests that a different phenomenon happened during the microfluidization of OBC-Sol-M. Probably some aggregation of small molecules happened during the microfluidization process, as observed by the bi-modal particle size and aggregates observed in the light microscopy of the OBC-Sol-M (Figure 3f). Low-Mw β-glucans (<200 kDa) have a high probability of forming aggregates and gel [32]. OBC-Sol-M had the lowest β-glucan Mw (148 kDa), so the formation of a network structure with the small β-glucan molecules could explain the smaller extractability and higher WRC of the OBC-Sol-M sample.

### 3.2. Properties of OBC Samples in Suspension

The viscosity of aqueous oat suspensions is affected by many factors, such as the temperature, particle size, concentration, solubility and Mw of various compounds (starch, β-glucan, protein, etc.) present in the suspension. In this study, after the initial phase of 60 min of heating at 50 °C, the highest viscosity was observed for the OBC suspension (305 mPa·s) followed by the OBC-Hyd suspension (198 mPa·s; Figure 4). These samples had a considerably larger particle size and β-glucan Mw than the other measured suspensions. Interestingly, the viscosity of these two samples increased as a function of time at 50 °C (Figure 4), while the other samples showed constant viscosity. This increase is probably due to the release and solubilization of β-glucan (and other compounds) from the large solid particles present in OBC and OBC-Hyd, whereas in the other samples that were more degraded by microfluidization or in soluble form (OBC-Sol), β-glucan was already released from the matrix and solubilized immediately. During heating to 90 °C, the viscosity of OBC and OBC-Hyd first decreased, but a considerable increase in viscosity was observed after the sample reached 90 °C, most likely due to starch gelatinization. The viscosity increase at 90 °C was less pronounced in the OBC-Hyd sample, possibly due to partial gelatinization of starch in the enzyme inactivation process (extrusion). Expectedly, the OBC-Sol suspension had the lowest viscosity, about 10 times smaller than the OBC suspension, apparently due to its low β-glucan Mw and low content of native starch. It should also be noted that the dry matter content of the OBC-Sol suspension was less than half (2.2%) that of the other suspensions (5%), as all suspensions were prepared at a standardized β-glucan content of 1% (β-glucan content of OBC-Sol sample twice as high as that in the other OBC samples). In addition to starch gelatinization and pasting, the final viscosity of the OBC suspensions after heating and cooling was also assumed to be affected by the Mw of the β-glucan present in the suspensions. The viscosity of oat samples has previously been linked to the Mw of β-glucan [32], and also in this study the viscosity of OBC suspensions decreased with a decrease in β-glucan Mw. The contribution of proteins to the viscosity of the suspensions was probably less pronounced, due to the low solubility and high denaturation temperature (~110 °C) of the main oat protein fraction (globulins) [34].

Microfluidization clearly reduced the viscosity of all OBC suspensions (Figure 4). An interesting observation was the extensive decrease in viscosity of the OBC suspension after microfluidization, which indicated that in addition to the decrease in particle size and β-glucan Mw, starch was also heavily degraded by the treatment under high pressure and mechanical stress. It has been previously reported that some starches can be at least partially gelatinized by high-pressure homogenization [35,36] and that many starch types can be fully gelatinized by high hydrostatic pressure treatment even at room temperature [37,38]. Valoppi et al. [33] showed that microfluidization increased the viscosity of an insoluble oat fraction (from 5 to 46 mPa) and the authors suggested that it was due to the higher amount of soluble material after microfluidization. The same authors showed that when the soluble oat fraction was microfluidized, a reduction in viscosity was observed, which was attributed to a possible degradation of β-glucan Mw [33].

The colloidal stability of OBC and OBC-Hyd suspensions were the lowest, as a high sediment volume was observed after 24 h (16.3 and 15.4 mL, respectively). The stability of these OBC samples was still higher than that of wheat bran, which in the same conditions was completely sedimented after 2 min [26]. Probably the higher amounts of insoluble particles and DF (such as arabinoxylan) in wheat bran compared to oat bran are behind the difference [39]. Compared to enzyme treatment, microfluidized OBC samples were better dispersed and their stability was considerably improved, as only 0.5–1.3 mL of sediment was measured after 24 h (Table 1). Microfluidization has been previously also shown to improve the stability of wheat bran, peeled wheat bran, wheat aleurone preparation and insoluble oat fraction [26,28,33]. The higher stability of the microfluidized OBC samples is presumably due to the smaller and more uniform particles that can better stay in suspension, but also to the higher amounts of soluble DF components generated by microfluidization [26,28]. For example, OBC is known to contain arabinoxylan [40] that could have been solubilized by the treatment [39].

### 3.3. Properties of Acid Milk Gels with OBC Samples

When the properties of the acid milk gels were studied, it was found that while control acid milk gels had 8.5% of spontaneous syneresis, none of the gels with added OBC samples showed spontaneous syneresis (data not shown). Control acid milk gels had 25.4% of WHC, which was considerably increased by the addition of oat samples by 3.3-, 2.3- and 1.8-fold for OBC, OBC-Hyd, and OBC-Sol, respectively (Figure 5a). Microfluidized OBC samples also improved the WHC compared to the control acid milk gel, but to a lower extent than the corresponding samples without microfluidization (Figure 5a). Concerning the firmness of the milk gels, the enrichment with the OBC samples made the gels softer compared to the control (Figure 5b). Milk gels enriched with microfluidized OBC samples had higher firmness than the corresponding samples without microfluidization when analyzed as freshly made, but the firmness was still lower than that of the control. When the acid milk gels were stored for 7 days, the texture of all gels became firmer, except for the OBC-Sol-M-enriched gel (Figure 5b). In addition, the firmness of the gel containing OBC-Sol clearly increased during storage, and it was the firmest gel among the OBC-containing samples, getting close to the gel sample without OBC.

The microstructure of the acid milk gels is shown in Figure 6. Compared to the control (Figure 6g), the gel enriched with OBC (Figure 6a) and OBC-Hyd (Figure 6b) had clear aggregates. These aggregates formed in the gel enriched with OBC (Figure 6a) were dense and non-homogeneous, forming large void areas (shown in black). Addition of microfluidized samples (OBC-M and OBC-Hyd-M) did not seem to disrupt the structure of the acid milk gels as much as OBC and OBC-Hyd, and β-glucan seemed to be more uniformly distributed in their gel structure (Figure 6d,e). Acid milk gel with OBC-Sol or OBC-Sol-M (Figure 6e,f) did not have as many aggregates as the other OBC samples, but the protein matrix had still an uneven distribution when compared to the control (Figure 6g).

There are few previous studies on the effects of oat or barley β-glucan addition on technological properties of dairy matrices (yogurt or acid milk gels) [15,16,17,18,22]. These studies using highly purified β-glucan provide good background information, but in the present study, less refined oat ingredients were evaluated in an attempt to provide new information for dairy industries interested in developing oat-rich acidified products. The improved WHC and reduced spontaneous syneresis by the addition of OBC samples to acid milk gels was also observed upon the addition of barley β-glucan in stirred yogurts [15]. The improvement of water entrapment in the network by oat DF samples has been generally attributed to the gel-like characteristics of β-glucan, but also to their high viscosity. Acid milk gel enriched with OBC had the highest WHC, which is probably due to the high viscosity of OBC and the presence of native starch. High water retention of the OBC enriched gel also led to a soft and smooth gel. The formation of weaker gels is probably due to the phase separation between oat β-glucan and milk protein, reducing the protein aggregation during acidification [17]. It is evident from the microscopic pictures that OBC clearly interacted with casein aggregates (Figure 6). Therefore, OBC might have entrapped water and colloidal particles in its highly viscous network, stabilizing the system and avoiding depletion effects [41]. OBC-Hyd, which had lower viscosity than OBC, was also able to hold more water than the control gels, but not as much as the OBC. Lazaridou et al. [16] studied the impact of the Mw of purified barley β-glucan Mw on the physical properties of acidified skim milk gels. They also observed that increasing the β-glucan Mw from 70 to 250 kDa decreased the syneresis of the gels, which was attributed to the high viscosity that could entrap the colloidal particles [16]. In this work, a positive correlation was found between the β-glucan Mw and water holding capacity of acid milk gels enriched with OBC samples (r = 0.87 *). However, it should be pointed out that OBC samples contain other compounds besides β-glucan (starch, protein and other types of fiber) that might have also impacted the properties of acid milk gels. The OBC-Sol sample seemed to have a smaller impact on acid milk gel characteristics than OBC or OBC-Hyd, most likely due to its soluble nature as compared with the more complex, partly insoluble matrix of OBC and OBC-Hyd.

## 4. Conclusions

In an attempt to improve the performance of oat bran in high-moisture food applications at addition levels enabling the health claim of β-glucan, enzymatic hydrolysis and microfluidization were investigated. Both β-glucanase treatment and microfluidization decreased the viscosity and improved the colloidal stability of the OBC samples. Microfluidization of a complex food raw material such as oat bran was shown for the first time to decrease the Mw of β-glucan, which would be a fast and efficient alternative to enzymatic hydrolysis required to reduce the viscosity and sliminess of oat bran in suspension. Microfluidization had an even greater impact than enzymatic treatment. These properties of OBC in suspension were linked with the reduction of the β-glucan Mw and particle size and the more homogeneous microstructure of the OBC particles caused by processing. Both treatments also increased the extractability of the soluble compounds from the OBC samples and increased the WRC of OBC, except when the soluble OBC fraction OBC-Sol was treated with microfluidization. The addition of all OBC samples to acid milk gels decreased syneresis and improved the water holding capacity compared to the control gels without OBC addition. The acid milk gels enriched with OBC samples were also softer than the control gel. The water holding capacity of the acid milk gels decreased with the reduction of β-glucan Mw of the OBC samples. The texture of the gels seemed to be less affected by microfluidized samples, as the gels were firmer than the ones enriched with enzyme-treated OBC samples (i.e., closer to the control gel without OBC samples). The findings of this study encourage the utilization of oat bran/whole grain oats in the development of β-glucan-rich high-moisture food products.

## Figures and Tables

**Figure 1 foods-11-00228-f001:**
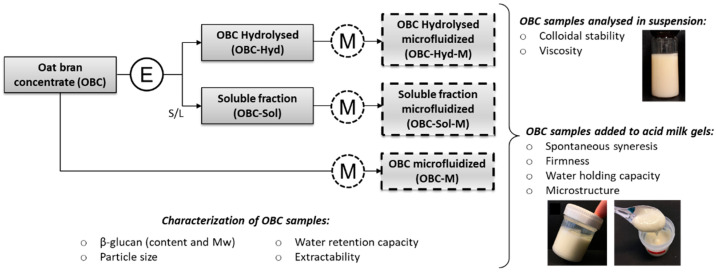
Setup of the experimental design. E = enzymatic treatment; M = Microfluidization; S/L = solid-liquid separation to recover the soluble fraction.

**Figure 2 foods-11-00228-f002:**
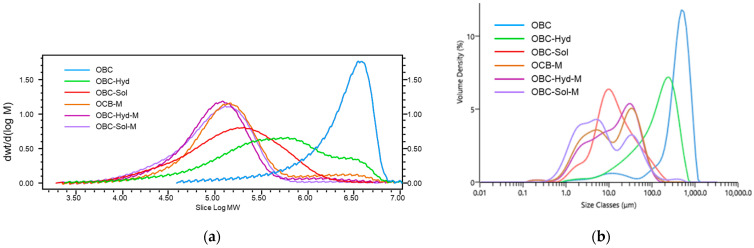
β-Glucan molecular weight distribution (Mw) (**a**) and particle size distribution (**b**) of OBC samples of OBC samples. OBC: Oat bran concentrate; OBC-Hyd: OBC hydrolyzed with β-glucanase; OBC-Sol: water-soluble fraction from OBC after hydrolysis with β-glucanase; OBC-M: OBC microfluidized; OBC-Hyd-M: OBC-Hyd microfluidized; OBC-Sol-M: OBC-Sol microfluidized.

**Figure 3 foods-11-00228-f003:**
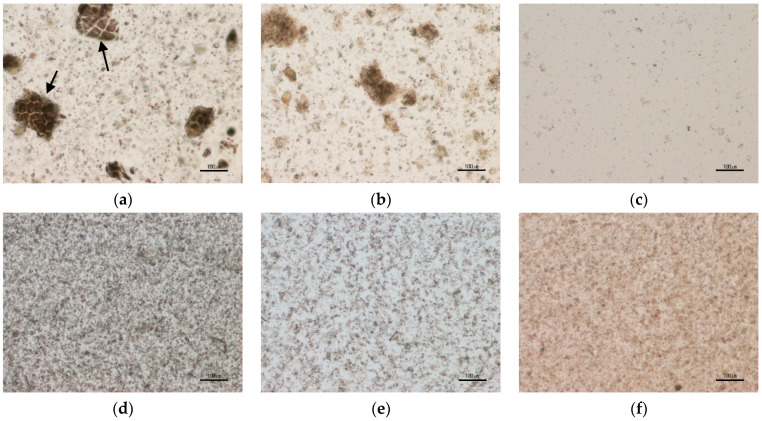
Microscopy of OBC samples in suspension: OBC (**a**), OBC-Hyd (**b**), OBC-Sol (**c**), OBC-M (**d**), OBC-Hyd-M (**e**), OBC-Sol-M (**f**). OBC: Oat bran concentrate; OBC-Hyd: OBC hydrolyzed with β-glucanase; OBC-Sol: water-soluble fraction from OBC after hydrolysis with β-glucanase; OBC-M: OBC microfluidized; OBC-Hyd-M: OBC-Hyd microfluidized; OBC-Sol-M: OBC-Sol microfluidized.

**Figure 4 foods-11-00228-f004:**
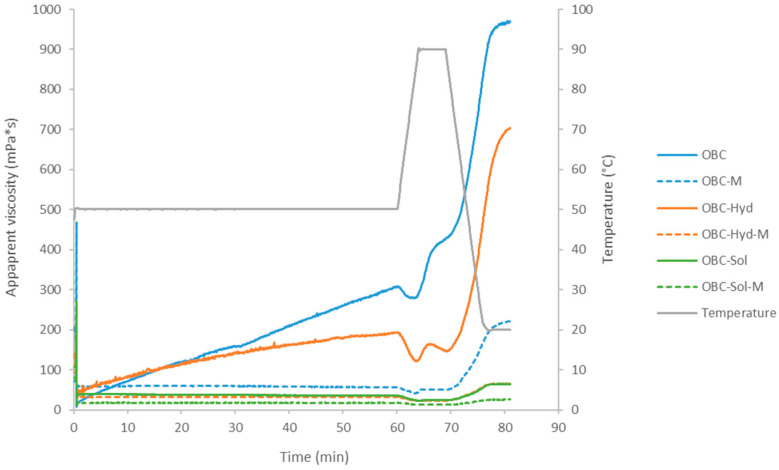
Viscosity (mPa·s) of the OBC samples in suspension as a function of time (min) and temperature (°C). OBC: Oat bran concentrate; OBC-Hyd: OBC hydrolyzed with β-glucanase; OBC-Sol: water-soluble fraction from OBC after hydrolysis with β-glucanase; OBC-M: OBC microfluidized; OBC-Hyd-M: OBC-Hyd microfluidized; OBC-Sol-M: OBC-Sol microfluidized.

**Figure 5 foods-11-00228-f005:**
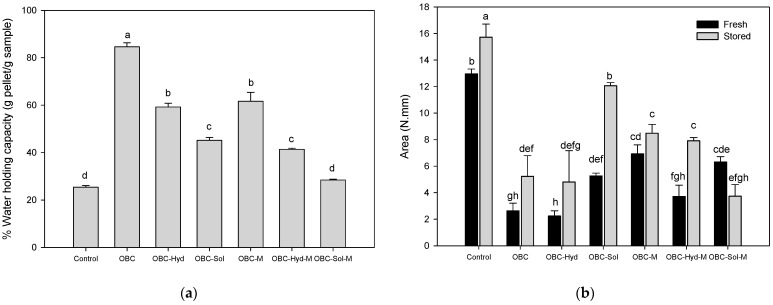
Water holding capacity (g pellet/g sample, %) (**a**) and firmness (area, N.mm) (fresh and 7 days stored) (**b**) of acid milk gels enriched with OBC samples and the control sample. Control = acid milk gel prepared only with skim milk powder (no oat samples added). Different letters (a–h) indicate statistically significant differences (*p* < 0.05) between the samples. OBC: Oat bran concentrate; OBC-Hyd: OBC hydrolyzed with β-glucanase; OBC-Sol: water-soluble fraction from OBC after hydrolysis with β-glucanase; OBC-M: OBC microfluidized; OBC-Hyd-M: OBC-Hyd microfluidized; OBC-Sol-M: OBC-Sol microfluidized.

**Figure 6 foods-11-00228-f006:**
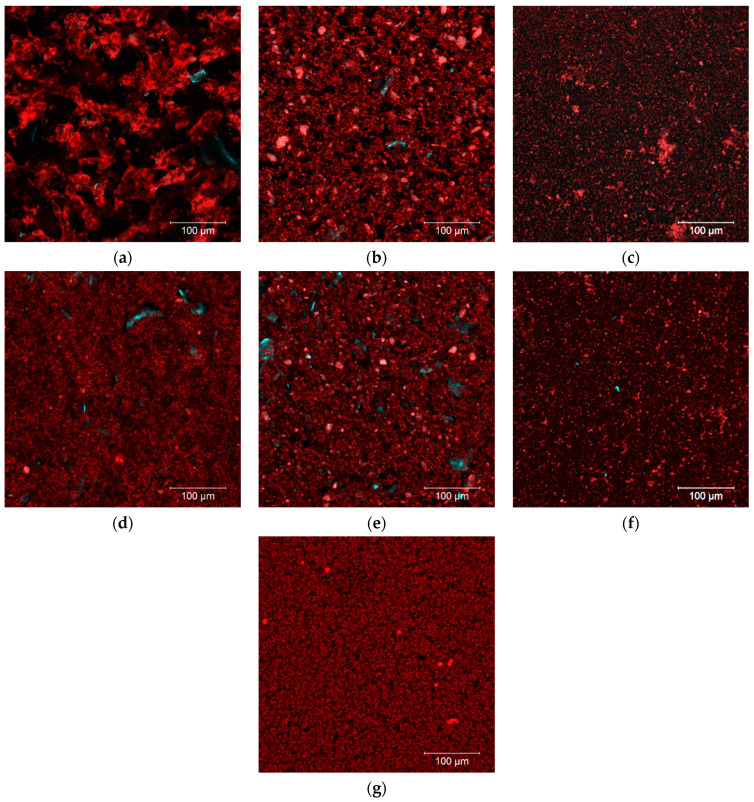
CLSM images of acid milk gels enriched with OBC (**a**), OBC-Hyd (**b**), OBC-Sol (**c**), OBC-M (**d**), OBC-Hyd-M (**e**), OBC-Sol-M (**f**) and the control acid milk gel (without any oat sample) (**g**). Proteins and fiber are stained with Rhodamine B (appears red) and Calcofluor White (appears cyan), respectively. The brightness of (**c**,**f**) was adjusted for better visualization of the protein matrix. OBC: Oat bran concentrate; OBC-Hyd: OBC hydrolyzed with β-glucanase; OBC-Sol: water-soluble fraction from OBC after hydrolysis with β-glucanase; OBC-M: OBC microfluidized; OBC-Hyd-M: OBC-Hyd microfluidized; OBC-Sol-M: OBC-Sol microfluidized.

**Table 1 foods-11-00228-t001:** Characteristics of OBC samples: β-glucan content, average particle size, WRC, extractability and properties in suspension.

	Β-Glucan			^3^ d_50_ (µm)	^4^ WRC (mL/g)	Extractability (%)	Apparent Viscosity (mPa·s)		Colloidal Stability (mL)		
	Content (g/100 g d.m.)	^1^ Mw (kDa)	^2^ Đ				At 60 min	Final	1 h	4 h	24 h
OBC	21.1 ± 0.2 ^b^	2748 ± 65	2.1 ± 0.2	426 ± 6 ^a^	3.0 ± 0.0 ^b^	9.9 ± 0 ^f^	305 ± 2 ^a^	969 ± 7 ^a^	1.5 ± 0.6 ^a^	2.3 ± 1.3 ^a^	16.3 ± 0.4 ^a^
OBC-Hyd	21.2 ± 0.2 ^b^	893 ± 42	6.7 ± 1.0	124 ± 5 ^b^	3.1 ± 0.1 ^b^	21.3 ± 1 ^e^	198 ± 21 ^b^	684 ± 22 ^b^	0.8 ± 0.3 ^ab^	1.5 ± 0.4 ^ab^	15.4 ± 1.3 ^a^
OBC-Sol	44.5 ± 0.8 ^a^	294 ± 20	4.1 ± 0.3	14 ± 0.4 ^c^	1.9 ± 0.2 ^d^	80.2 ± 1 ^a^	34 ± 2 ^cd^	63 ± 4 ^d^	1.0 ± 0.8 ^ab^	1.4 ± 0.8 ^ab^	4.0 ± 2.9 ^b^
OBC-M	n.a.	350 ± 2	4.4 ± 0.5	12 ± 0.2 ^c^	2.5 ± 0.2 ^c^	27.3 ± 2 ^d^	56 ± 2 ^c^	218 ± 5 ^c^	0.5 ± 0.0 ^b^	0.5 ± 0.0 ^b^	0.5 ± 0.0 ^c^
OBC-Hyd-M	n.a.	203 ± 10	3.2 ± 0.1	15 ± 0.2 ^c^	2.3 ± 0.1 ^c^	47.6 ± 1 ^b^	32 ± 1 ^cd^	65 ± 2 ^d^	0.8 ± 0.3 ^ab^	0.6 ± 0.3 ^b^	1.3 ± 0.6 ^bc^
OBC-Sol-M	n.a.	148 ± 9	2.7 ± 0.2	6.2 ± 0.1 ^c^	5.8 ± 0.1 ^a^	38.9 ± 0 ^c^	17 ± 1 ^d^	26 ± 1 ^e^	1.0 ± 0.0 ^ab^	1.1 ± 0.2 ^ab^	1.1 ± 0.2 ^bc^

^1^ Mw = weight average molecular weight, ^2^ Đ = dispersity (Mw/Mn), ^3^ d_50_ = average particle size, ^4^ WRC = water retention capacity, n.a. = not analyzed. Different letters (a–f) within each column indicate statistically significant differences (*p* < 0.05) between the samples. OBC: Oat bran concentrate; OBC-Hyd: OBC hydrolyzed with β-glucanase; OBC-Sol: water-soluble fraction from OBC after hydrolysis with β-glucanase; OBC-M: OBC microfluidized; OBC-Hyd-M: OBC-Hyd microfluidized; OBC-Sol-M: OBC-Sol microfluidized.

## Data Availability

The data presented in this study are available on request from the corresponding author. The data are not publicly available due to agreement of the project.

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
