# Peer review of "Impact of Enzymatic Hydrolysis and Microfluidization on the Techno-Functionality of Oat Bran in Suspension and Acid Milk Gel Models"

_foods, 2022, doi:10.3390/foods11020228_

Round 1
Reviewer 1 Report
In this study, oat bran concentrate (OBC) was treated with enzymatic hydrolysis, physical treatment, and the combination of the two. In addition, the physicochemical properties of OBC with an attempt to improve OBC behavior in high-moisture food applications was investigated. Experiments were laid out and results were analyzed. However, there are some mistakes and problems in the manuscript which should be well checked and revised.
- Microfluidization should be briefly introduced in the introduction.
- Refine references in all methods.
- Line144-145:“OBC suspensions (5% concentration prepared as described in the item 2.5.1)”. There is no 2.5.1 in your paper and reference [25], please check it.
- Preparation of 5% suspensions was described in 2.4, 2.5, 2.6 and 2.7.1. Are these suspensions prepared consistently? If so, why 2.7.1 and 2.7.2 were placed under the title "Characterization of OBC samples in Suspension" of 2.7, while 2.4, 2.5 and 2.6 were not included? It is suggested to write the preparation methods of 5% suspensions at the beginning of this section.
- What is the active β-glucan content in 5% suspension? Do you want to keep the fixed β-glucan content per sample?
- Line 227-228: You mentioned the “Microfluidization of the OBC samples significantly reduced β-glucan Mw, also after the enzyme treatment (Table 1)”. So, it suggests to provide a significantly analysis in Table 1.
- Line 244-245:“positive correlation was found between the β-glucan Mw and the average particle size (d50) of OBC samples (r =0.99**)”. How is the positive correlation (r =0.99**) obtained?And the same as Line 288, Line 410.
- Line 222:The OBC-Sol sample with Mw (297 kDa),but it was 294 in Table 1.
- The unit of WRC is mL/g or ml/g in Table 1? And it should be mL?
- Line 241:“at an OBC concentration of 5% (i.e., 1.1, 1.1 and 2.2% of β-glucan for OBC-M, OBC-Hyd-M and OBC-Sol-M, respectively)”. The β-glucan content of the OBC-M, OBC-Hyd-M and OBC-Sol-M was not measured in Table 1, so I don't understand how this result was calculated.
- Figure 2 provides only the scale, but not the magnification.
- Line 248-249: “OBC presented intact structure of the outer layers, such as pericarp and rows of aleurone layer (Figure 2a)”. It is recommended that related structures should be marked in the image.
- RVA is usually programmed in accordance with the starch gelatinization process. The starch concentration in your sample is very low. Is it reasonable to use RVA to measure viscosity?
- It is suggested that the appendix figure be reasonably integrated into the text, because the relevant indicators are very important.
- Why is the protein matrix in Figure 5c, f so significantly different from the others? You did not provide more detail about the results of Figure 5c, f.
Reviewer 2 Report
Manuscript Number: Foods-1525770
Title: Impact of enzymatic hydrolysis and microfluidization on the techno-functionality of oat bran in suspension and acid milk gel models
Overview and general recommendation
The article structure is compact, sequential and logical. The data are adequate to support the conclusion. The methods section provides sufficient information on design, sampling, definitions, data collection and data analysis. References are up-dated adequate and correctly cited.
Minor comments:
- I recommend that in the whole manuscript to be reviewed the writing of the units of measurement (for example, ml or mL?)
- To clarify the Materials and Methods formula for calculating water retention capacity (or water holding capacity – Figure 4?).
- I consider that the part of the Conclusions can be improved with concrete results about the research carried out.
The authors carry out an interesting work and I recommend it for publication.

Reviewer 3 Report
Manuscript is written in readable manner. But following points have to be addressed?
- Why bran concentrate (OBC) hydrolyzed with with beta-glucanase?
- In Material and Methods: Section 2.2. Provide block diagrams for the enzymatic treatment and microfluidization?
- Microfluidization should be described in more details?
- There are mechanical methods to increase Beta-glucan content in oat. Please provide a comparison of those methods with your methods used to show viability in the industrial scale?
Reviewer 4 Report
Review on manuscript: foods-1525770
Impact of enzymatic hydrolysis and microfluidization on the techno-functionality of oat bran in suspension and acid milk gel models
by Natalia Rosa-Sibakov, Maria Julia de Oliveira Carvalho, Martina Lille and Emilia Nordlund
submitted to Foods
In the manuscript submitted for comments, the authors studied the effect of hydrolysis and microfluidization on the functionality of oat bran.
In my opinion, the manuscript is interesting and prepared correctly. It also has an application potential. However, the authors should try to answer the question whether the degradation of oat glucans, improving their functional properties, also causes a decrease in biological activity? The weakness of the work is the lack of a statistical evaluation of the obtained results.
Detailed recommendation:
lines 76-82 – the literature review should end with a clearly formulated goal resulting there from, the presentation of the methods used is not necessary here,
line 102 – does the deactivation of enzymes under such conditions not result in additional glucans degradation?
line 109 – condition of freeze drying should be specified,
line 120 – should be: β,
line 122 – model, producer and origin country of HP SEC system should be mentioned, basic information on the conditions of the analysis should be given,
line 125 – what references were used?
line 137 – model, producer and origin country should be mentioned,
lines 157-158 – this dependence can be expressed by the equation,
line 173 – what parameters were determined?
line 244 – information about the calculation of correlation coefficients should be added to the methodology in the section statistical analysis,
Table 1 – data should be evaluated statistically,
Table 1, and next, Figure 3 – it can be assumed that the analyzed solutions are non-Newtonian liquids, therefore the apparent viscosity will be a more appropriate description,
Figure 4 – data should be evaluated statistically,
Figures A1 and A2 – may be included in the main text of the manuscript.
Round 2
Reviewer 3 Report
Please re-do the checking of errors in writing and typo.
Author Response
Dear reviewer,
We have re-done the checking and spell-checked the whole manuscript to correct typos and grammar. The revisions are marked in red font.